# Host Cell and SARS-CoV-2-Associated Molecular Structures and Factors as Potential Therapeutic Targets

**DOI:** 10.3390/cells10092427

**Published:** 2021-09-15

**Authors:** Jitendra Kumar Chaudhary, Rohitash Yadav, Pankaj Kumar Chaudhary, Anurag Maurya, Rakesh Roshan, Faizul Azam, Jyoti Mehta, Shailendra Handu, Ramasare Prasad, Neeraj Jain, Avaneesh Kumar Pandey, Puneet Dhamija

**Affiliations:** 1Department of Zoology, Shivaji College, University of Delhi, New Delhi 110027, India; jnujitendra@gmail.com (J.K.C.); rakesh.biotech85@gmail.com (R.R.); 2Department of Pharmacology, All India Institute of Medical Sciences (AIIMS), Rishikesh 249201, India; shailendra.handu@gmail.com; 3Molecular Biology & Proteomics Laboratory, Department of Biotechnology, Indian Institute of Technology (IIT), Roorkee 247667, India; pkchaudharydu@gmail.com (P.K.C.); ramasare.prasad@bt.iitr.ac.in (R.P.); 4Department of Botany, Shivaji College, University of Delhi, New Delhi 110027, India; anuragvns.maurya@gmail.com; 5Department of Pharmaceutical Chemistry and Pharmacognosy, Unaizah College of Pharmacy, Qassim University, Unaizah 51911, Saudi Arabia; faizulazam@gmail.com; 6Department of Physiology, Maharishi Markandeshwar Medical College and Hospital, Solan 173229, India; jiajyoti@gmail.com; 7Department of Cancer Biology, CSIR-Central Drug Research Institute, Lucknow 226031, India; neeraj.monc@aiimsrishikesh.edu.in; 8Department of Pharmacology, Postgraduate Institute of Medical Education and Research, Chandigarh 160012, India; avaneeshkpandey@gmail.com

**Keywords:** SARS-CoV-2, coronavirus disease 19, pathogenesis, therapeutic targeting, angiotensin-converting enzyme 2

## Abstract

Coronavirus disease 19 (COVID-19) is caused by an enveloped, positive-sense, single-stranded RNA virus, referred to as severe acute respiratory syndrome coronavirus 2 (SARS-CoV-2), which belongs to the realm *Riboviria*, order *Nidovirales*, family *Coronaviridae*, genus *Betacoronavirus* and the species *Severe acute respiratory syndrome-related coronavirus*. This viral disease is characterized by a myriad of varying symptoms, such as pyrexia, cough, hemoptysis, dyspnoea, diarrhea, muscle soreness, dysosmia, lymphopenia and dysgeusia amongst others. The virus mainly infects humans, various other mammals, avian species and some other companion livestock. SARS-CoV-2 cellular entry is primarily accomplished by molecular interaction between the virus’s spike (S) protein and the host cell surface receptor, angiotensin-converting enzyme 2 (ACE2), although other host cell-associated receptors/factors, such as neuropilin 1 (NRP-1) and neuropilin 2 (NRP-2), C-type lectin receptors (CLRs), as well as proteases such as TMPRSS2 (transmembrane serine protease 2) and furin, might also play a crucial role in infection, tropism, pathogenesis and clinical outcome. Furthermore, several structural and non-structural proteins of the virus themselves are very critical in determining the clinical outcome following infection. Considering such critical role(s) of the abovementioned host cell receptors, associated proteases/factors and virus structural/non-structural proteins (NSPs), it may be quite prudent to therapeutically target them through a multipronged clinical regimen to combat the disease.

## 1. Introduction

Severe acute respiratory syndrome coronavirus 2 (SARS-CoV-2) causes highly transmissible coronavirus disease 19 (COVID-19) with a myriad of varying symptoms and a spectrum of disease severity, ranging from asymptomatic to critical illness [1]. The initial outbreak of the disease was reported in Wuhan, China, in late 2019 [2], and later on, it spread far and wide, covering almost every national and international territory, and geographical boundary, thereby causing unprecedented global socioeconomic disruption, psychosomatic anomalies, innumerable mortality and unimaginable suffering [3]. SARS-CoV-2, as of 4:50 pm CEST, 1 September 2021, has reportedly caused around 217 million laboratory-confirmed infections and 4.5 million deaths worldwide (https://covid19.who.int/; accessed on 2 September 2021). This virus enters the human body, primarily through nasal and oral passages, and then gains cellular entry via molecular interaction between its glycosylated homotrimeric structural spike (S) protein and host cell membrane-bound cognate receptor, angiotensin-converting enzyme 2 (ACE2). Therefore, the tissue expression and distribution of the ACE2 receptor directly influence host range, viral tropism and pathogenesis [4]. In fact, any internal or external/environmental factors, leading to upregulation of ACE2 receptor expression, may serve as risk factors for severe COVID-19. For instance, Smith et al. found an increase in ACE2 expression in the respiratory tract following exposure to cigarette smoke and inflammatory signals, suggesting a higher susceptibility of such individuals towards severe COVID-19 [5]. The ACE2 receptor is known to have also been used by previously reported human coronaviruses, such as NL63 and SARS-CoV. Historically, there have been several reports of outbreaks of mild upper-respiratory illness-causing coronaviruses, including human coronaviruses (hCoVs)-OC43, -HKU, -NL63, and -229E [6], however, they have not been as devastating and debilitating as current outbreak-causing SARS-CoV-2. These four categories of hCoVs account for 15–30% of cases of non-fatal common cold in adult humans, although they may cause fatal lower-respiratory tract infection in immunocompromised persons, elderly people and certain infants [7]. In contrast, over the recent past, we have witnessed outbreaks of highly evolved and pathogenic human coronaviruses, such as 2002–2003 SARS-CoV and 2012 MERS-CoV, with death rates of around 10% and 36%, respectively. Unlike SARS-CoV’s dependence on ACE2 receptor, MERS-CoV relies upon dipeptidyl peptidase 4 (DPP4) as the cell entry receptor [8]. Most recently, an outbreak of a novel coronavirus was reported in late 2019, initially called 2019-nCoV, but later renamed as SARS-CoV-2 by the International Committee on Taxonomy of Viruses (ICTV) on 11 February 2020, following pre-set standard guidelines developed by the Food and Agriculture Organization of the United Nations (FAO) and the World Organisation for Animal Health (OIE) (https://www.who.int/emergencies/diseases/novel-coronavirus-2019/technical guidance/naming-the-coronavirus-disease-(covid-2019)-and-the-virus-that-causes-it; accessed on 5 August 2021).

Compared with both SARS-CoV (also referred to as SARS-CoV-1) and MERS-CoV, SARS-CoV-2 is thought to be considerably less fatal but highly contagious, leading to worldwide spread of infection and, as a consequence, the occurrence of an once-in-a-century pandemic, as the whole world has currently been witnessing. The virus may follow various routes of transmission, such as respiratory droplets, fecal-oral, mother-to-baby (also called as vertical transmission), sexual and ocular route [9]. Among them, respiratory-droplet mode of transmission is the most common. The continuance of the current pandemic has overwhelmed the already stretched thin healthcare facility, especially in low- and middle- income countries (LMICs), along with substantial disruption in socioeconomic growth and development [10]. Furthermore, there have been considerable loss to the teaching and learning processes globally owing to closure of colleges and universities due to imposition of lockdown, with underdeveloped and developing countries being the most affected ones.

SARS-CoV-2 gains access to host cell machinery following occurrence of successful interaction between its S protein and host cell ACE2 receptor, culminating in cellular invasion. The SARS-CoV-2 spike (S) protein, a homotrimeric class I fusion glycoprotein, consists of two functionally distict parts, namely S1 and S2. The S1 part possesses a receptor-binding domain (RBD) that helps the virus in engaging with the host cell receptor, whereas, the S2 domain with its hepatad repeat (HR) regions and fusion peptide mediates the fusion of the viral membrane with the host cell membrane [11,12]. This molecular interaction between the virus spike protein and ACE2 receptor is facilitated by synchronized working of various protease molecules/factors, including TMPRSS2 (transmembrane serine protease 2) and furin [13]. The SARS-CoV S/ACE2 interface has been well studied at the atomic level, which potentially indicates that the degree of compatibility between the S protein and the ACE2 receptor positively correlates with virus transmissibility [14]. Therefore, inhibition of S protein–ACE2 interaction using specific pharmaceutical molecules and/or a specific antibody may prove quite effective against the contagion as a prospective antiviral intervention. Furthermore, SARS-CoV-2 may also use receptors, such as neurophilin 1 (NRP-1), neurophilin 2 (NRP-2) and C-type lectin receptors (CLRs). Neurophilin 1 is abundantly expressed in respiratory and olfactory epithelium, and has been shown to bind furin-cleaved substrates, such as S1, thereby significantly enhancing the SARS-CoV-2 infection potential. For instance, Cantuti-Castlevetri L. et al. categorically showed the presence of SARS-CoV-2-infected NRP-1-positive cells through co-immunostaining of olfactory epithelium obtained from human COVID-19 autopsies. Therefore, such potentiation of infectivity can be considerably reduced and/or blocked by using either RNA interference, selective inhibitors, monoclonal anti-NRP-1 antibodies, or a combination thereof, suggesting NRP-1 as a potential therapeutic target [15,16,17] (Figure 1).

Irrespective of the receptor(s) utilized and molecular mechanism(s) deployed by SARS-CoV-2 for cellular entry, once inside the host cell, the virus releases its linear, single- stranded 5′-capped and 3′poly-A-tailed 30 kb RNA genome. This is followed by the onset of a complex mechanism of viral gene expression under tight spatio-temporal regulation. The 30 kb viral genome eventually encodes around 14 open reading frames (ORFs), resulting in translational synthesis of an array of structural and non-structural proteins, which are capable of performing a myriad of structural, molecular and enzymatic functions, as well as rebuilding virus progenies for new infection [18,19]. Two overlapping ORFs, ORF1a and ORF1b, located immediately downstream of 5′-UTR (untranslated region, 265 nucleotides long) and occupying 2/3 of the total genomic length, encode continuous polypeptides pp1a and pp1ab, respectively. The pp1ab forms owing to a programmed -1 ribosomal frameshift at the very short overlap of both ORF1a and ORF1b [20]. The intrinsic differential efficiency of the ribosomal frameshift mechanism corresponds to a specific stoichiometry between pp1a and pp1ab, wherein pp1a is expressed 1.4–2.2 folds higher as compared to pp1ab. Thereafter, both polyproteins (pp1a and pp1ab) undergo auto-proteolytic processing owing to two intrinsic cysteine proteases, namely papain like protease (PL^PRO^) located within NSP3, and chymotrypsin-like protease located within NSP5, thereby forming 16 non-structural proteins (NSP1-16). In general, the protease activity contained within NSP5 is also referred to as 3C-like protease (3CL^PRO^) and/or main protease (M^PRO^) owing to its resemblance to picornaviral 3C protease and its major contribution towards cleaving the majority of polyprotein cleavage sites, respectively. NSP1, also deemed to be the host shutoff factor, is proteolitically released quite rapidly, and then specifically targets the host cell translation machinery by sterically occluding the mRNA channel in ribosomal subunits, such as 40S and 43S pre-initiation complex, as well as non-translating 80S ribosomes. As a consequence, the cytosolic functional ribosomal pool is drastically reduced, allowing translation of highly efficient viral mRNAs over less efficient cellular 5′ UTRs. This results into hijacking of the host cell translation machinery by SARS-CoV-2, as observed in cases of other viral infections as well [21]. NSP2–NSP16 follow a specific pattern of 3-dimensional molecular association, resulting in multiprotein functional enzyme molecules, called the replicase/transcriptase complex (RTC), which occupies a specific subcellular location, and eventually decides the course of the replication cycle [22]. Further, while NSP2–NSP11 provide support for accommodation of viral RTC along with modulation of intracellular membranes and host immune evasion, NSP12–NSP16 possess core enzymatic functions, such as RNA synthesis, proofreading, and requisite modification essential for completion of the normal virus cycle [23]. The role of each non-structural protein (NSP) is well defined individually and combinatorially as follows: NSP3 (papain-like protease for polyprotein processing, de-ADP-ribosylation, deubiquitination, interferon antagonist, and double-membrane vesicle (DMV) formation along with NSP4 and NSP6), NSP5 (main protease, interferon signaling inhibitor), NSP7–NSP8 (primase complex, cofactor for RdRp), NSP9 (binding of single-stranded RNA), NSP10 (cofactor for NSP14 and NSP16), NSP12 (nucelotidyl transferase, primary RNA-dependent RNA polymerase, abbreviated as RdRp), NSP13 (helicase/triphosphatase), NSP14 (3′ to 5′ exoribonuclease, proofreading, RNA 5′-cap formation, guanosine N7-methyltransferase), NSP15 (an endonuclease, host immune evasion) and NSP10/NSP14/NSP16 (N7- and 2′O-methyltransferases, RNA cap formation) [19,24,25]. Notwithstanding consistent scientific efforts, the functional roles of NSP2 and NSP11 have not yet been revealed. Furthermore, the remaining 1/3 genomic region lying downstream of ORF1a/ORF1b is predicted to have 13 ORFs, capable of encoding various proteins, such as four structural proteins- spike (S), membrane (M), envelope (E), and nucleocapsid (N); and 9–10 putative accessory proteins/factors [25]. Apart from their direct functional involvement at multiple levels in the production of viral progenies, these SARS-CoV-2 proteins have also been found to interact with a myriad of host cell proteins involved in various cellular pathways, as well as modulation of gene expression through epigenetic and non-epigenetic regulation. The extent of SARS-CoV-2–host cell protein–protein interactions coupled with modulation of gene expression may also play an important role in pathogenesis and clinical outcome. For instance, a recent empirical work based on affinity purification mass spectrometry (AP-MS) has shown 332 highly reliable SARS-CoV-2–human protein–protein interactions (PPIs), suggesting the likelihood of a global multipronged alteration in cellular and biochemical mechanisms, such as nuclear transport, lipoprotein metabolism, and biogenesis of the ribonucleoprotein complex, following viral infection [26]. Such a substantial global alteration in cellular pathways may be the underlying reason for COVID-19 intractable etiology, prognosis and pathogenesis. However, such a broad spectrum of molecular interactions may, at least some if not all, serve as potential therapeutic targets as well. In fact, such premise has led to the identification of multiple druggable host proteins and/or factors [26], which may result in successful repositioning and repurposing of FDA-approved drugs, as well as discovery of novel drugs for COVID-19 therapy [25,27]. Here, we mainly focus on possible druggable targets on both host cells and SARS-CoV-2, whose in-depth understanding are prerequisite to design multipronged treatment modalities to combat the disease.

## 2. Prospective Therapeutic Targets

So far, there has not been any major breakthrough with respect to the development of a specific and effective treatment regimen for COVID-19. Therefore, an incessant global quest is on to find potential druggable targets, which could either be host-specific, virus-specific, or both. Although some success has been achieved in terms of the development of vaccines and antiviral drugs, nevertheless, effective and specific treatment is still awaited and urgently required. Here, we make an effort to discuss some potential prospective molecular targets on both virus and host cells, which are being considered for designing drugs and therapeutic antibodies.

### 2.1. Potential Therapeutic Targets on Host Cells

The SARS-CoV-2’s S protein engages with the host cell receptor, ACE2, to gain cellular entry, leading to onset of infection. The structural spike (S) protein contains multiple domains, including the N-terminal receptor-binding domain (RBD), and protease cleavage sites, such as the furin site [28]. The RBD of the S protein plays a very important role in molecular interaction, therefore, therapeutic approaches specifically targeting the RBD/ACE2 interface may prove to be one of the most reliable treatment modalities. For instance, empirical findings based on multidisciplinary approaches have zeroed in on some potential molecules/drugs, such as Evans blue, sodium lifitegrast, and lumacaftor, that may specifically interfere with the SARS-CoV-2-S–ACE2 interaction, thereby preventing infection and disease occurrence [29]. During this interaction, the S protein is specifically primed and cleaved at the S1/S2 cleavage site by activated host tissue-specific proteases, such as TMPRSS2, TMPRSS4, TMPRSS11D, and furin, facilitating virus internalization and genome release [13,30]. Moreover, endosomal proteases, such as cathepsin B (CTSB), cathepsin L (CTSL), basigin (BSG) and FURIN may also be potentially involved in this process, making them potential therapeutic targets [31]. Owing to the significant involvement of proteases, such as TMPRSS2 and furin in priming the S protein, which are prerequisite for cellular entry of the virus, they may also be targeted from a therapeutic point of view using a protease inhibitor, such as camostat mesylate. In fact, such approach, involving camostat mesylate, has proven to be considerably effective in blocking SARS-CoV-2 infection in primary human lung cells [13].

Apart from the ACE2 receptor, SARS-CoV-2, like many other viruses, such as human T-lymphotropic virus-1 (HTLV-1) [32] and Epstein–Barr virus (EBV) [33], also uses multifunctional transmembrane receptor neurophilin 1 (NRP-1) for cell entry. NRP-1 is known to be expressed abundantly in respiratory and olfactory epithelia, and has a broad range of implications owing to its involvement in various cellular processes, such as angiogenesis, axonal guidance, growth and progression, tumor progression, immune functions and viral entry [17,34]. Vascular endothelial growth factor A (VEGF-A) is considered to be one of the most important ligands of the neurophilin receptor, and is primarily involved in angiogenesis, but was recently discovered to be pro-nociceptive as well [35]. It is generally thought that SARS-CoV-2 leverages the VEGF-A interaction site on NRP-1 to gain cellular entry. The likelihood of usage of NRP-1 may be substantiated by the report of COVID-19 patients showing upregulation of the receptor in lung samples [17]. Therefore, successful blockade of the interaction between SARS-CoV-2 and NRP-1 by using well established inhibitors, such as EG00229 and EG01377 [36], may prove to be an effective COVID-19 therapy (Figure 1).

Furthermore, there have been efforts to identify host genes (pro-viral and anti-viral) essential for SARS-CoV-2 infection. Such tireless efforts aim at developing an understanding about the viral pathogenesis, as well as finding out novel therapeutic target(s). For example, Wei J. et al. carried out genome-wide CRISPR screening and identified multiple active host genes with crucial roles in histone modification and chromatin regulation, cellular signaling, and RNA regulation. Identification of active genes encoding the pleotropic HMGB1 protein and members of the SWI/SNF chromatin remodeling complex in SARS-CoV-1, SARS-CoV-2, and NL63-infected Vero-E6 cells unequivocally suggests a relation between epigenetic regulation and viral pathogenesis. HMGB1 was found to intrinsically regulate ACE2 expression, indicating the pivotal involvement of the epigenetic process in SARS-CoV cellular entry and infection, while a small-molecule antagonist inhibited the same in monkey and human cells, further substantiating the relevance of the epigenetic mechanism [37]. Therefore, developing greater insights into the underlying mechanism of such processes may help in screening/designing small therapeutic molecules, as well as repurposing of FDA-approved drugs to prevent infection and disease.

Apart from targeting the cell receptor, ACE2, and associated host factors/proteases, such as TMPRSS2, another therapeutic approach may entail targeting the SARS-CoV-2–cellular protein–protein interaction (interactome). A recent study, involving multiple expressed SARS-CoV-2 proteins as baits, identified 332 host cell interacting proteins (overlapping and specific), belonging to various functional categories and/or natures, which are generally involved in several complex biological processes and pathways (Figure 2). Of the total interacting proteins, around 40% of host cell proteins belong to the endomembrane compartment and/or vesicle trafficking pathways [26].

Interestingly, several SARS-CoV-2 proteins interact with innate immune signaling proteins. For instance, NSP13, NSP15, and Orf9b (also referred to as nonstructural protein NS9b) target the interferon (IFN) pathway by interacting with various pathway-associated proteins. Similarly, NSP13 and Orf9c (also referred to as nonstructural protein NS9c) can alter the NF-κB pathway, which is involved in multiple crucial cellular processes, including immune response. Further, NSP9 and Orf3a (also referred to as nonstructural protein NS3a) show molecular associations with antiviral immune signaling-associated E3 ubiquitin ligases, tripartite motif-containing protein 59 (TRIM59) and mind bomb 1 (MIB1), respectively [38,39]. SARS-CoV-2’s Orf6 (also referred to as nonstructural protein NS6) interacts with an IFN-inducible mRNA nuclear export complex, NUP98-RAE1, in a manner similar to vesicular stomatitis virus (VSV), influenza A, and polio virus amongst others, and thereby antagonizes interferon signaling through disruption at the level of nuclear export [40]. The nucleocapsid (N) protein of SARS-CoV-2 binds to multiple translational regulators, such as UPF1 and MOV10 (mRNA decay factors), CK2 (protein kinase) and host mRNA binding proteins. Furthermore, N protein can also influence 5′ cap-dependent translation, thereby negatively affecting global protein synthesis [41]. Around a dozen of SARS-CoV-2 proteins undergo Sec61 translocon-mediated cotranslational insertion into the endoplasmic reticulum (ER) and remain localized in the virus replication complex [42]. SARS-CoV-2 NSP8 interacts with three components of the signal recognition particle (SRP), thereby negatively influencing Sec61-mediated protein translocation into the ER, which may be effectively inhibited by PS3061, a Sec61 inhibitor. This inhibitor may also interfere with SARS-CoV-2 replication and assembly as observed in the case of other enveloped RNA viruses [43,44].

### 2.2. Potential Therapeutic Targets on SARS-CoV-2

SARS-CoV-2 primarily relies on the host cell receptor, ACE2, to gain successful cellular entry. Molecular interaction between the SARS-CoV-2’s S protein and ACE2 is accomplished by conformational changes, protease cleavage on the S1/S2 domain, and eventual fusion of viral membrane with host cell membrane [45]. The virus can follow either endocytic or non-endocytic pathways to enter the host cell, and eventually releases its nearly 30 kb genome in the cytosol. Thereafter, a cascade of molecular events leads to synthesis of virus structural and non-structural proteins and multiple copies of viral genomes, which eventually assemble together, forming new viral progenies [19]. Moreover, the S protein is quite immunogenic in nature, leading to the formation of a diverse set of antibodies and memory B cells, each specific to a particular epitope/region/domain of this multidomain structural protein. For instance, a recently published work, employing single-cell sorting, identified 453 neutralizing antibodies and 4277 SARS-CoV-2 spike protein-specific memory B cells from 14 COVID-19 convalescent individuals. This work also shows that there are differential neutralizing capabilities among these antibodies with anti-RBD being the most potent one, followed by the anti-S1 domain, anti-spike protein trimer, and the anti-S2 subunit being the least potent [46]. The most potent antibody may, either in its natural or engineered form, be used as a therapeutic and/or prophylactic measure, and delivered/administered to the patient through intravenous infusion [47,48]. The abovementioned facts are the underlying reason as to why most of the vaccines, irrespective of their type, mainly rely on the S protein as the active antigen ingredient. While choosing therapeutic antibodies, one must also factor in their efficiency and effectiveness against emerging variants of concern (VOC) containing D614G, E484K, and N501Y substitutions [49]. Such variants have been evolving and circulating globally throughout the COVID-19 pandemic, and continue to prevail and show upward infection trend, leading to thousands of death each day around the world. Their rapid global spread is attributed to the acquisition of higher infectivity as a result of favorable mutational changes in the virus genome, especially in the gene encoding S protein. Moreover, diverse antibodies have also been detected against other SARS-CoV-2 structural proteins, such as anti-E, anti-M, and anti-N with varied potential and therapeutic implications [50] (Figure 3). However, the quantities of such antibodies in COVID-19 patients are lower owing to the relatively small molecular sizes of these antigens (M, E, and N proteins), as well as the lesser degree of structural protrusion of their corresponding ectodomains, which are prerequisite for engagement and recognition by B cells and other immune cells. In addition to various structural proteins, non-structural proteins may also acts as antigens as evidenced by the presence of reactive CD4^+^ T cells against NSP3, NSP4, Orf3a (NS3a), and Orf8 (NS8). Considering all these factors, it may be prudent to design a vaccine, involving both categories of potential antigens, i.e., structural and non-structural proteins, as it may evoke a diverse set of B cells to make antibodies, thereby providing holistic protection against infection. A good vaccine must be able to induce both B and T cells along with formation of corresponding long-term memory cells with minimal or no side effects.

Apart from therapeutic antibodies, SARS-CoV-2-specific proteins and processes, favoring the infectious virus, may also be targeted using well established small molecules, such as aloxistatin, chloroquine/hydroxychloroquine, anti-viral nucleotide analogs (remdesivir), protease inhibitors (lopinavir and ritonavir), antiviral phytochemicals, and the broad-spectrum antiviral drugs like favipiravir and arbidol. In general, aloxistatin is a cysteine protease inhibitor for calpain and cathepsins, and used as a cancer therapy drug. Since cathepsin L has been reported to play a crucial role in SARS-CoV-2 cell entry as well [51], administration of aloxistatin may be very important in combating infection. Moreover, aloxistatin may also bind SARS-CoV-2 main protease (M^PRO^), as well as papin-like proteases, albeit with lower specificity, thereby interfering with the proteolysis of polypeptides 1a/ab [52]. Owing to the very high sequence specificity of M^PRO^, compounds structurally mimicking its substrate cleavage site may be very precise inhibitors with negligible or no adverse effect on host cellular proteases [53]. Furthermore, chloroquine and hydroxychloroquine are well established antimalarial drugs, and are being tested for COVID-19 therapy. Whereas chloroquine inhibits terminal phosphorylation of ACE2, hydroxychloroquine elevates endosomal pH, both being crucial processes prerequisite for successful establishment of SARS-CoV-2 infection. Owing to their involvement in such crucial cellular processes, several clinical trials are underway to establish the efficiency and modalities with respect to these drugs before final approval as candidate drugs against SARS-CoV-2 infection is granted (https://clinicaltrials.gov/; accessed on 6 August 2021). Similarly, RdRp and 3Clpro (also termed M^PRO^), highly conserved SARS-CoV-1/2 proteins, are also very specific targets to be employed for COVID-19 treatment. Remdesivir and ritonavir/lopinavir, ribonucleotide analogs, have also been found to be capable of interfering with the working of RdRp, and therefore constitute another set of effective candidate drugs against the current pandemic (Figure 3) [54,55,56]. Furthermore, several in silico analyses are also being carried out to find novel drugs and/or bioactive natural compounds to treat COVID-19 [57,58,59,60,61,62].

## 3. Conclusions

Notwithstanding consistent global efforts, there have not been yet any major successful development and availability of potential drugs to treat SARS-CoV-2-induced COVID-19. Therefore, it is quite urgent to work in this direction, involving multidisciplinary approaches and multinational collaboration, to tackle the current pandemic and save millions from death, socioeconomic and psychosomatic devastation. However, to accomplish such a grand feat, we first need to unequivocally focus our attention on prospective therapeutic target molecules on/in both host cells and the virus to design holistic therapy. In fact, potential target(s) on the host might preclude the challenges imposed owing to rapid mutation and, as a consequence, the development and evolution of therapeutically resistant strains, including those belonging to disconcerting categories like variant of concern (VOC) and variant of high consequence (VOHC) (https://www.who.int/en/activities/tracking-SARS-CoV-2-variants/; accessed on 3 September 2021). These variants have been originating and evolving worldwide, replacing the wild-type Wuhan strain (Wuhan-Hu-1; NC_045512), and are responsible for multiple waves of infection across the globe. Furthermore, therapeutic targeting of the host–virus interface would also offer similar advantages as such targets are less likely to undergo mutational resistance. Moreover, several drugs and small-molecule antagonists might target the epigenetic mechanism, thereby regulating the gene ecoding the ACE2 receptor. This would eventually deprive SARS-CoV-2 of its binding site on the host, and thereby helps in preventing infection. Considering the abovementioned facts and related variables, targeting the host and the host–virus interface, as well as drugs developed for such a purpose, may be very specific, efficient and durable.

On the other hand, therapeutic targeting may involve SARS-CoV-2-specific molecules, factors (various structural proteins, particulary the S protein), and mechanisms, including cellular entry, proteolysis of PP1a/ab, viral genome transcription/replication, and viral assembly. SARS-CoV-2 possesses several types of structural and non-structural proteins with specific intrinsic molecular conformations and functional characteristics, which may prove to be quite specific targets and help in designing clinical and therapeutic modalities. For instance, most of the vaccines, either developed and approved or under development, specifically target the antigenic spike (S) protein, which then evokes adaptive immune cells, producing antibodies and corresponding memory B and T cells as part of a preventive and prophylactic strategies. Moreover, drugs are also being developed to target virus-specific mechanisms and enzymes. For instance, a clinical trial involving the repurposed broad-spectrum antiviral drug remdesivir, a nucleoside analog capable of interfering with the working of RNA-dependent RNA polymerase (RdRP), is underway to find out its efficacy against COVID-19 [50]. Furthermore, several drugs, including camostat mesylate, might interfere at the level of cellular entry, thereby preventing infection. Last but not least, we need to be very cautious, spread awareness among communities, and follow COVID-appropriate behavior until the pandemic is over.

## Figures and Tables

**Figure 1 cells-10-02427-f001:**
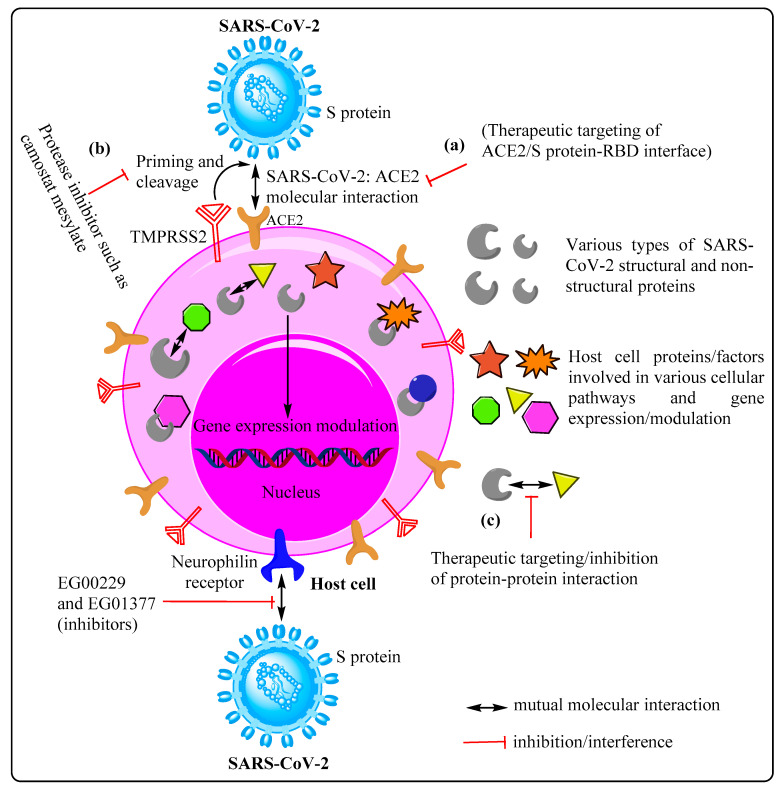
Host cell receptor/molecules/factors as therapeutic targets for FDA-approved drug repurposing. Broadly, therapeutic approaches can be designed by keeping in consideration three categories of host cell molecules: (**a**) receptor(s) such as ACE2 and neurophilin (NPR) facilitating cellular entry of SARS-CoV-2, (**b**) associated factors/proteases mediating receptor priming and cleavage, and (**c**) SARS-CoV-2: host cell protein–protein interaction (interactome).

**Figure 2 cells-10-02427-f002:**
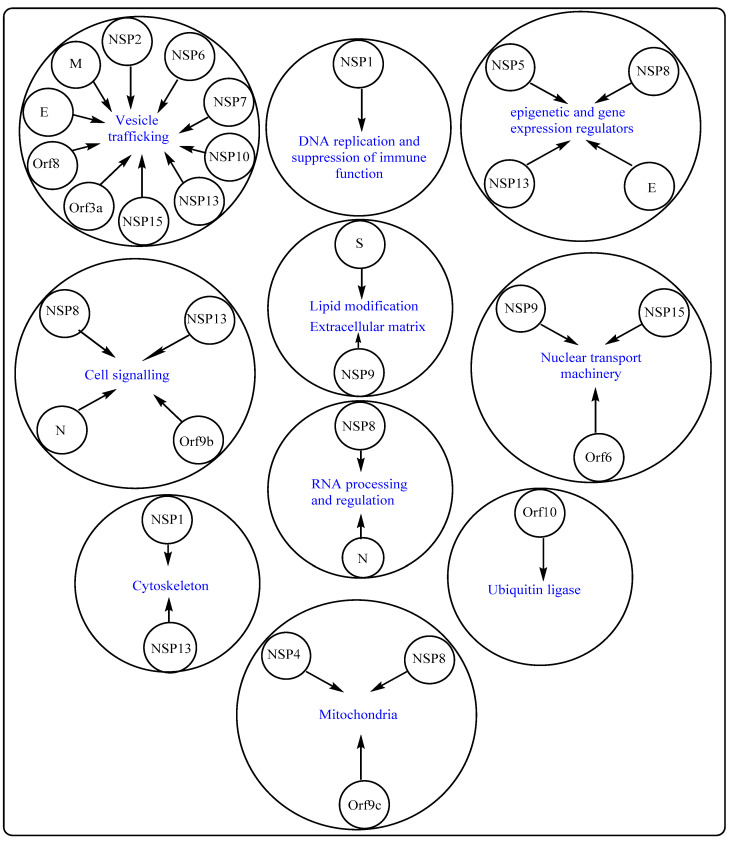
Interactome, involving SARS-CoV-2 proteins and cellular proteins. SARS-CoV-2 structural and non-structural proteins have been found to interact with multiple host cell proteins involved in various cellular processes, as well as remain associated with several cell organelles. SARS-CoV-2 proteins are shown in the center of the small circle, whereas human host cell proteins are placed in the center of the large circle. An arrow indicates a possible interaction, which may or may not have significant implications. N-Neucleocapsid protein, S-Spike protein, M-Membrane protein and E-Envelope protein.

**Figure 3 cells-10-02427-f003:**
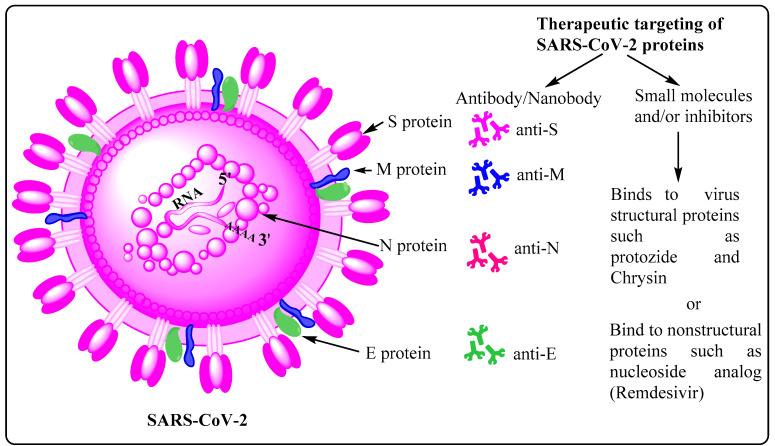
Potential targets on SARS-CoV-2. SARS-CoV-2, the causative organism of COVID-19, possesses four types of structural proteins, namely spike (S), membrane (M), envelope (E), and nucleocapsid (N). Of these, the homotrimeric S protein shows the highest immunogenicity, leading to the production of correspondingly high amount of the anti-S antibody, and generation of memory B and T cells. Anti-M, anti-N and anti-E antibodies, albeit in lesser quantities, are also detected in samples derived from COVID-19 patients.

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
