# Peer review of "Host Cell and SARS-CoV-2-Associated Molecular Structures and Factors as Potential Therapeutic Targets"

_cells, 2021, doi:10.3390/cells10092427_

Round 1

Reviewer 1 Report

The paper entitle “Host Cell and SARS-CoV-2-Associated Molecular Structures  and Factors as Potential Therapeutic Targets” is a very interesting and very usefully review, take into account that the actually pandemic situation, about the mechanism of SARS-CoV-2 entry in the cells and the medicines that actually are being used to avoid the SARS-CoV-2 infection. Moreover the author gives us information about the mechanism of action of these drugs shown the inhibitory effect of each one.

However some minor questions may be answered.

  • The authors indicate that the virus is capable of using different receptors (ACE2, Neurophilin1 or 2) for entry into the cell. Is it known if the effect is different depending on whether you use one or the other? is the virulence of the virus the same?
  • In both sections potential therapeutic targets on host cells and potential therapeutic targets on SARS-CoV-2, it would be useful to include, in addition to the figure, a table with each of the therapeutic targets showing the currently used therapies.
  • In figure 2 the immune system is not included however this system is one of the system targeted by the virus.
  • In line 231 author say that some drugs as Remdesivir and ritonavir/lopinavir, ribonucleotide analogues, are capable of interfering with the working of RdRp, and make reference to figure 3 however this is not showed in figure 3.

Author Response

Authors’ Response to Reviewers 1’ Comments and Suggestions

Comments and Suggestions for Authors

The paper entitle “Host Cell and SARS-CoV-2-Associated Molecular Structures  and Factors as Potential Therapeutic Targets” is a very interesting and very usefully review, take into account that the actually pandemic situation, about the mechanism of SARS-CoV-2 entry in the cells and the medicines that actually are being used to avoid the SARS-CoV-2 infection. Moreover the author gives us information about the mechanism of action of these drugs shown the inhibitory effect of each one.

 However some minor questions may be answered.

  • The authors indicate that the virus is capable of using different receptors (ACE2, Neurophilin1 or 2) for. Is it known if the effect is different depending on whether you use one or the other? is the virulence entry into the cell of the virus the same?

Response: First of all, authors express their collective gratitude towards reviewer for invaluable suggestion and comments.

It is well know that host cell surface receptor (ACE2 and NPR1/2) expression and distribution positively correlate with host selection, viral tropism and pathogenesis. Furthermore, NRP1 is known to potentiate the overall SARS-CoV-2 infectivity, however, it is not yet revealed whether the virulence, disease severity and clinical outcome based on receptor-specificity/usage would be drastically different (Science. 2020 Nov 13; 370(6518): 856–860) (Science 2020:Vol. 370, Issue 6518, pp. 861-865).

  • In both sections potential therapeutic targets on host cells and potential therapeutic targets on SARS-CoV-2, it would be useful to include, in addition to the figure, a table with each of the therapeutic targets showing the currently used therapies.

Response: The content with respect to drugs/therapeutic targets have been improved and updated.

  • In figure 2 the immune system is not included however this system is one of the system targeted by the virus.

Response: Suppression of host immune system by Nsp1 has been added in Fig 2.

  • In line 231 author say that some drugs as Remdesivir and ritonavir/lopinavir, ribonucleotide analogues, are capable of interfering with the working of RdRp, and make reference to figure 3 however this is not showed in figure 3.

Response: It has been added under small molecules in fig 3.

Reviewer 2 Report

In my opinion your review is short, but it is quite good organized and comprehensively described.

There are readable Figures, what is big for plus to You.

I am clinician, that's why for me it could be more information about nowadays used drugs.

Also it would be interesting write a little bit more about currently ongoing research about mentioned molecular structures and factors.

In the section 2.2 there are few  connected words such as "measuresand" or "deliveredto". They should be disconnected.

Author Response

Authors’ Response to Reviewers 2’ Comments and Suggestions

Comments and Suggestions for Authors

  • In my opinion your review is short, but it is quite good organized and comprehensively described.

There are readable Figures, what is big for plus to You.

I am clinician, that's why for me it could be more information about nowadays used drugs.

Also it would be interesting write a little bit more about currently ongoing research about mentioned molecular structures and factors.

Response: First and foremost, authors express their gratitude towards your time and effort for reviewing and providing your invaluable suggestions with regard to the manuscript. Keeping comments in mind, there have been revisions in this regard to improve, elaborate and update the overall content in the light of emerging knowledge.

  • In the section 2.2 there are few connected words such as "measuresand" or "deliveredto". They should be disconnected.

Response: Connected words have been rectified as per the reviewer’s suggestion.